# Strong positive selection biases identity-by-descent-based inferences of recent demography and population structure in *Plasmodium falciparum*

Bing Guo [1,2], Victor Borda [1], Roland Laboulaye[1], Michele D. Spring[3], Mariusz Wojnarski[3], Brian A. Vesely [3], Joana C. Silva [1,4,5], Norman C. Waters[3], Timothy D. O'Connor [1,6] ✉ & Shannon Takala-Harrison [2,6] ✉

Malaria genomic surveillance often estimates parasite genetic relatedness using metrics such as Identity-By-Decent (IBD), yet strong positive selection stemming from antimalarial drug resistance or other interventions may bias IBD-based estimates. In this study, we use simulations, a true IBD inference algorithm, and empirical data sets from different malaria transmission settings to investigate the extent of this bias and explore potential correction strategies. We analyze whole genome sequence data generated from 640 new and 3089 publicly available *Plasmodium falciparum* clinical isolates. We demonstrate that positive selection distorts IBD distributions, leading to underestimated effective population size and blurred population structure. Additionally, we discover that the removal of IBD peak regions partially restores the accuracy of IBD-based inferences, with this effect contingent on the population's background genetic relatedness and extent of inbreeding. Consequently, we advocate for selection correction for parasite populations undergoing strong, recent positive selection, particularly in high malaria transmission settings.

Malaria, a mosquito-borne disease caused by *Plasmodium* parasites, is a leading cause of illness and death in many developing countries, with an estimated 247 million cases and 619,000 malaria-related deaths in 2021[1]. *Plasmodium falciparum* (*Pf*) is responsible for most malaria cases and deaths. Antimalarial drugs have imposed one of the strongest selective pressures on the *Pf* genome, with the parasite having evolved resistance to nearly all drugs used as first-line therapies[2–5]. Eastern Southeast Asia (SEA) has historically been an epicenter of emerging antimalarial drug resistance, leading to intensive malaria control efforts in this geographic region that have resulted in an 80%

decrease in malaria incidence over the last twenty years[1]. The decline in *Pf* incidence in SEA is accompanied by a structured *Pf* population, as well as decreased genetic diversity or effective population size ($N_e$)[6–9]. Monitoring the dynamics of *Pf* demography, population structure, and gene flow is a critical component of malaria surveillance efforts to inform targeted elimination intervention strategies and prevent the spread of drug-resistant parasites[10].

In population genetics, Identity-By-Descent (IBD) is a highly informative metric used to estimate $N_e$[11–13] and fine-scale population structure for recent generations[14–17]. An IBD segment is a continuous

[1]Institute for Genome Sciences, University of Maryland School of Medicine, Baltimore, MD, USA. [2]Center for Vaccine Development and Global Health, University of Maryland School of Medicine, Baltimore, MD, USA. [3]Armed Forces Research Institute of Medical Sciences, Bangkok, Thailand. [4]Department of Microbiology and Immunology, University of Maryland School of Medicine, Baltimore, MD, USA. [5]Global Health and Tropical Medicine (GHTM), Instituto de Higiene e Medicina Tropical (IHMT), Universidade NOVA de Lisboa (NOVA), Lisbon, Portugal. [6]These authors jointly supervised this work: Timothy D. O'Connor, Shannon Takala-Harrison. ✉e-mail: timothydoconnor@gmail.com; STakala@som.umaryland.edu

genomic region over which a pair of isolates or genomes share an identical sequence inherited from their most recent common ancestor (MRCA) without being broken down by recombination[18–20]. In selectively neutral scenarios, the length and positional distribution of IBD segments, as well as pairwise and population-level aggregates, can provide valuable insights into the recent evolutionary history of a population, enabling the estimation of $N_e$[12,21], population structure, genetic relatedness[9], and migration[6,22] in a time-specific manner[23,24]. Although many IBD-based inference tools were initially developed for human studies, they have recently been applied to malaria parasites[6,9,21,25,26], despite the large differences in evolutionary parameter values between humans and $Pf$ parasites. These differences include strong selection coefficients[27], high recombination rates (but comparable mutation rate)[28], and declining $N_e$[1] in $Pf$ compared to humans. Such differences can potentially affect the quality of IBD segment detection and the patterns of IBD sharing, and thus must be considered when applying IBD-based analysis in $Pf$.

Indeed, $Pf$ parasites have been under strong selection pressure due to intense malaria control efforts in recent decades, especially the widespread use of antimalarial drugs[29–32], resulting in the rapid emergence and spread of multidrug-resistant parasites in SEA and other malaria-endemic areas[3]. For instance, parasites in this region have developed resistance to chloroquine, the antifolates, mefloquine, and more recently, the artemisinin derivatives and their partner drugs[33,34]. Haplotypes harboring mutations conferring drug resistance have undergone strong selective sweeps with high selection coefficients (0.03–0.32)[27,35], multiple orders of magnitude greater than those usually observed in the human genome, where selection coefficients are often on the order of 0.001[36]. Such selective sweeps can bias inference based on IBD by changing the distribution of IBD segments and their aggregates. Positive selection increases IBD sharing at the locus under selection as well as neutral loci linked to the selected locus (genetic hitchhiking)[37], leading to an increase in linkage disequilibrium (LD)[38], long haplotypes, and long IBD segments[37,39]. The shift of the IBD distribution has therefore been used as a signal to detect selection in both humans[37,40] and malaria parasite populations[9,41]. Given the known effects of positive selection on IBD sharing patterns in human genomes[37], it is critical to understand whether and how IBD-based analysis is biased by positive selection in $Pf$, where selection coefficients are substantially greater than in humans.

However, the evaluation of potential selection bias in $Pf$ is complicated by the high genetic relatedness of parasites and inbreeding in lower malaria transmission settings, such as SEA, where there has been a rapid decline of the parasite population. This high background (genome-wide) LD, and IBD sharing[6,9,42,43] as a result of inbreeding and the low effective recombination rate, can bury the genomically local signal of positive selection, making it difficult to locate or correct.

Also complicating evaluation of potential selection bias is the high recombination rate in the $Pf$ genome (including both effective and actual rate)[41,44,45]. Although the mutation rates in $Pf$ and humans are comparable[28], the recombination rate is 60–70 times higher in $Pf$[46]. The relatively low ratio of mutation to recombination rate in $Pf$ (if ignoring background selection) leads to a small number of variants per genetic unit (low marker density), which is known to impact the ability to accurately detect IBD segments and to bias the IBD distribution[20]. Given these factors, we need a context-specific evaluation of the effect of positive selection on IBD-based inferences of demography in $Pf$.

In this study, we employed population genetic simulations and genealogy-based true IBD segments to evaluate how positive selection, with varying parameters, affects the IBD distribution and IBD-based estimates of $N_e$ and population structure in $Pf$. We proposed heuristic strategies to detect and remove genomic regions with excess IBD due to recent positive selection (IBD peaks) and evaluated whether the removal of IBD peaks mitigates positive selection-induced bias in the IBD distribution, $N_e$ estimation and population structure inference.

We then validated the findings from simulation analyses in empirical whole-genome sequencing (WGS) data sets from low and high malaria transmission settings.

## Results

### Parasite isolates and WGS data summary

To investigate the impact of positive selection on the inference of $N_e$ and population structure, we mainly focused on eastern SEA, as it has been a hotspot for drug resistance emergence[3,47]. We analyzed WGS data from 2055 $Pf$ isolates that passed quality control and data processing filters (see "Methods"), including 751 (640 new) isolates from Cambodia and Thailand that were sequenced in-house and 1304 eastern SEA isolates from the publicly available MalariaGEN Catalogue of Genetic Variation in $P. falciparum$ v6.0 ($Pf$6)[48]. The included isolates are distributed across 14 years and 18 provinces in four countries (Cambodia, Thailand, Laos, and Vietnam) (Fig. 1a). Among these isolates, 79.3%, 68.0%, and 46.1% isolates had at least 5x, 10x, and 25x coverage over >80% of the $Pf$ genome, respectively (Fig. 1b). The $F_{ws}$ statistic was estimated for each isolate that passed genotype missingness filtering to identify monoclonal versus polyclonal isolates, with 80% being classified as monoclonal isolates ($F_{ws}$ > 0.95) (Fig. 1c). Among the polyclonal isolates, 44.3% harbored a predominant clone (defined in "Methods"), and the predominant haploid genomes of these isolates (Fig. 1d, with ratio < 1.0), along with the phased haploid genomes of the monoclonal isolates, were included in the analyzable data set. Isolates from West Africa (WAF, $n$ = 1674 analyzable isolates) were also obtained from the MalariaGEN $Pf$6 database for validation of results in a high transmission setting (Supplementary Fig. 1). Among WAF isolates that passed quality control, 50.7% were monoclonal, consistent with the higher multiplicity of infection (MOI) expected in a high malaria transmission setting[28] (Supplementary Fig. 1c).

### Effects of positive selection on IBD distribution and IBD-based $N_e$ inference

Empirical data sets sampled from a real $Pf$ population often deviate from an ideal population due to various evolutionary factors, such as declining malaria incidence[1], parasite population structuring[7,8,49], selective sweeps[3,27,34], and asymmetrical gene flow or migration[6]. To evaluate the direct effect of positive selection on demographic inference, we conducted population genetic simulations using simplified models that reflect parameter values observed in $Pf$, such as strong positive selection, decreasing $N_e$, and a high recombination rate. We designed two categories of models: (1) a single-population model to test the effects of selection on the IBD distribution and the estimation of effective population size, and (2) a multiple-population model to test the effects of positive selection on IBD-based population structure inference.

To prevent the confounding of low-quality IBD calls on the evaluation of selection effects, we implemented a true IBD inference algorithm called tskibd. The algorithm directly utilizes ancestral information from simulated (true) genealogical trees in tree sequence format[50–52] and avoids phased genotype-based IBD inference (Supplementary Fig. 2a). We verified the quality of true IBD by comparing IBD-based $N_e$ estimates (via IBDNe[12]) with the true population size in neutral simulations under different demographic scenarios (Supplementary Fig. 2b).

Different aspects of the IBD distribution represent distinct types of information about evolutionary histories, such as the time to most recent common ancestor (TMRCA, inferred based on IBD segment length)[22,24], genetic relatedness or population structure (inferred based on pairwise genome-wide total IBD)[14,16,17,53], and selection detection (inferred based on IBD positional enrichment)[9,37,41]. We used the single-population simulation model to generate genetic data under neutral (Supplementary Fig. 3a) and other selection scenarios (Supplementary Fig. 3b–j) consistent with realistic evolutionary parameters for $Pf$ (see "Methods"). We found that strong positive

selection impacts multiple aspects of the IBD distribution, including increasing the proportion of longer IBD segments (Fig. 2a) and isolate-pairs sharing larger genome-wide total IBD (Fig. 2b) and enriching IBD around selected sites (Fig. 2c). More importantly, we found that $N_e$ (via IBDNe[12]) is underestimated in recent generations in cases with selection compared to neutral cases (Fig. 2d), likely due to the

increase in longer IBD segments (thus smaller $N_e$ in more recent time frames).

We evaluated IBD peak region identification and removal strategies to test whether positive selection-induced bias can be corrected (Supplementary Fig. 4 and "Methods"). In brief, peaks were identified on each chromosome using a threshold method, then validated

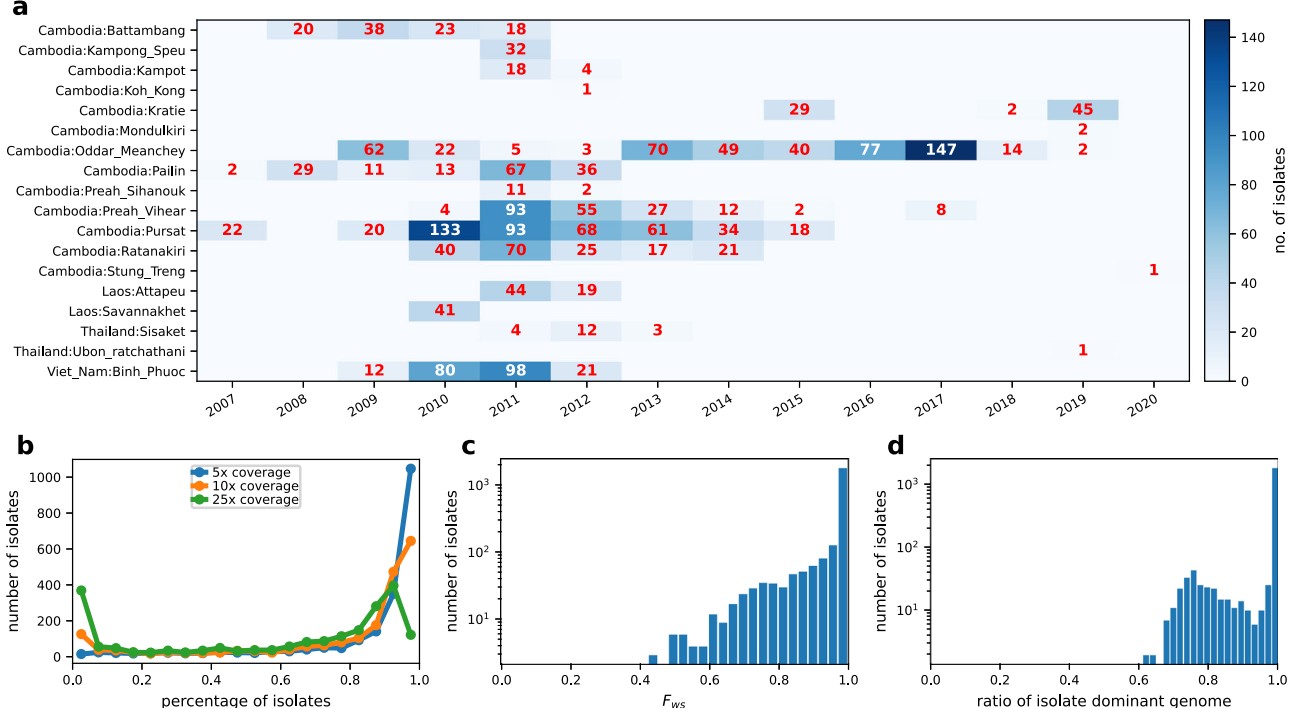

**Fig. 1 | Summary of *Pf* parasite isolates and WGS data from SEA. a** Distribution of sampling location and collection year for the 2055 analyzable samples. The text and color in each block indicate the number of isolates sampled at a given year from a given location (also see colorbar). **b** Distribution of genome fractions covered by at least 5, 10, and 25 sequence reads of all analyzable parasite genomes from SEA. **c** Distribution of $F_{ws}$ in sequenced isolates that passed genotype missingness

filtering. Note that to obtain a more accurate distribution of $F_{ws}$, polyclonal isolates without a predominant clone were included in this analysis. **d** Distribution of ratios of predominant haploid genomes (clones) in analyzable SEA isolates. The predominant clone of a polyclonal infection was determined by dEploid[90,92]. Source data are provided as a Source data file.

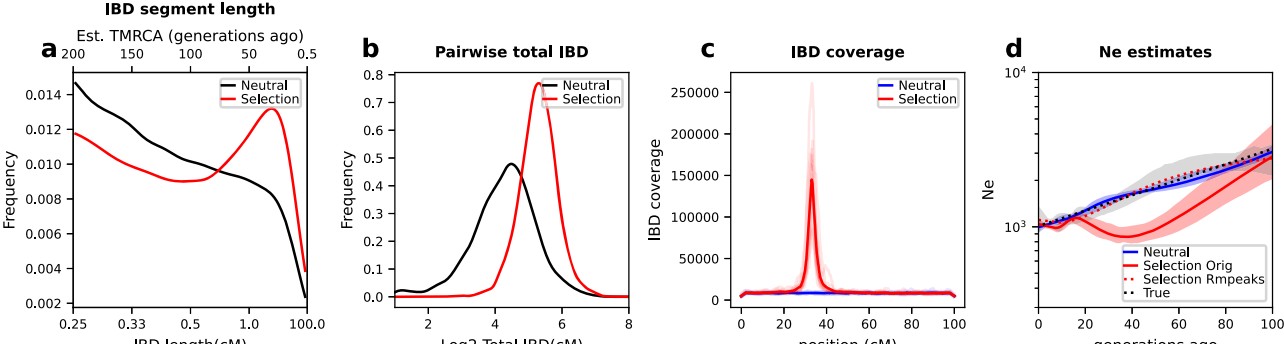

**Fig. 2 | Effects of positive selection on IBD distribution and $N_e$ inference.**
**a**–**c** Positive selection affects various aspects of the IBD distribution, including IBD segment length (**a**), total IBD shared by a pair of isolates (**b**), and IBD location along the chromosome (**c**). Note: (1) the x-axis in (**a**) uses a custom scale for IBD length $L$ (bottom) so that the estimated TMRCA ($50/L$, top) is in a linear scale; (2) for IBD segment length distribution analysis, shorter IBD segments (0.2–2 centimorgan (cM)) were included to cover the more distant past (>25 generations ago). Lines of transparent colors in (**c**) represent IBD coverage for different chromosomes for the same genome set; lines of solid colors show the average across chromosomes. The representative results were generated using a selection coefficient, $s$, of 0.3, a selection starting time 80 generations ago, and a single origin of the favored allele

introduced at the position of 33.3 cM of each chromosome. **d** Strong positive selection causes underestimation of $N_e$ compared to neutral simulation. The difference between selection ($s = 0.3$, red solid line) and neutral (blue solid line) scenarios can be partially mitigated by removing IBD segments (red dotted line) located within IBD peak regions. Parameter true population size (black dotted line) is plotted for reference. Error bands indicate 95% confidence intervals as determined by IBDNe[12]. Abbreviations: Neutral, neutral simulation; Selection (Orig), positive selection with IBD peak regions not removed; Selection Rmpeaks, positive selection with IBD peak regions removed. Source data are provided as a Source data file. For results for different selection parameter values, see Supplementary Fig. 3.

through integrated haplotype score[54] (iHS)-based selection statistics $X_{iHS}$ (see "Methods"). IBD segments located within validated peaks that have a large, local impact on the genome, as estimated by the peak impact index (defined in "Methods" and Supplementary Note 1), were removed. We found that removing IBD peak regions corrects the $N_e$ estimation in the selection scenario, mimicking the neutral $N_e$ estimates and true population size (Fig. 2d).

We further evaluated the impact of varying selection parameters, including selection coefficients (Supplementary Fig. 3b–d), selection starting times (Supplementary Fig. 3e–g), and the number of origins of the favored alleles (Supplementary Fig. 3h–j), on IBD distribution and $N_e$ estimates. In general, stronger selection (Supplementary Fig. 3d), intermediate selection duration time (Supplementary Fig. 3f), and a small number of origins (such as a hard sweep) (Supplementary Fig. 3h) allow the establishment of the selective sweeps (the favored allele is not lost during the sweep) (Supplementary Fig. 3 first column) and thus result in selection bias. Signed-rank tests based on replicated simulations suggested the effects of positive selection on $N_e$ estimates are statistically significant (Bonferroni-adjusted $p$ values < 0.05) (Supplementary Fig. 5).

## Effects of positive selection on population structure inference

Given the pronounced effects of positive selection on the IBD distribution and $N_e$ inference, it is vital to understand its impact on the inference of population structure. We assessed this impact using a multi-deme, one-dimensional stepping-stone model[55] with spreading selective sweeps (Fig. 3a) that simulates a pattern of allele frequency gradients across subpopulations (Fig. 3b), mimicking selective sweeps in a structured parasite population[56].

Under the neutral scenario with a moderate migration rate (such as 0.01, corresponding to 1% of individuals in a subpopulation being migrants from adjacent subpopulations in each generation), within-population IBD sharing dominates the pairwise sharing heatmap (Fig. 3c [left panel], and d [black line]), the total population is highly modular[57,58] with respect to the true subpopulation labels (Fig. 3e [left bar]), and community-detection using the InfoMap clustering algorithm[58,59] captures the true population structure with high consistency (Fig. 3f [left panel]).

However, with strong selection, both within- and between-population IBD sharing increases (Fig. 3c [middle panel]). This change results in an elevated ratio of inter-population to intra-population IBD sharing (Fig. 3d [blue line]), reduced network modularity (Fig. 3e [middle bar]), and collapsed community groups (Fig. 3f [middle panel]), making it difficult to distinguish one population from adjacent populations. We observed consistent patterns across varying selection strengths (Supplementary Fig. 6) and repeated simulations (Supplementary Table 1), suggesting the blurring effect is selection strength-dependent.

The effect of selection on structure inference can be partially mitigated by removing IBD segments located within genomic regions harboring IBD peaks (Supplementary Fig. 4). After the selection correction, the dominance of within-population IBD sharing and the modularity of population is restored (Fig. 3c [right panel], d [dashed red line], and e [right bar]), and the collapsed communities become

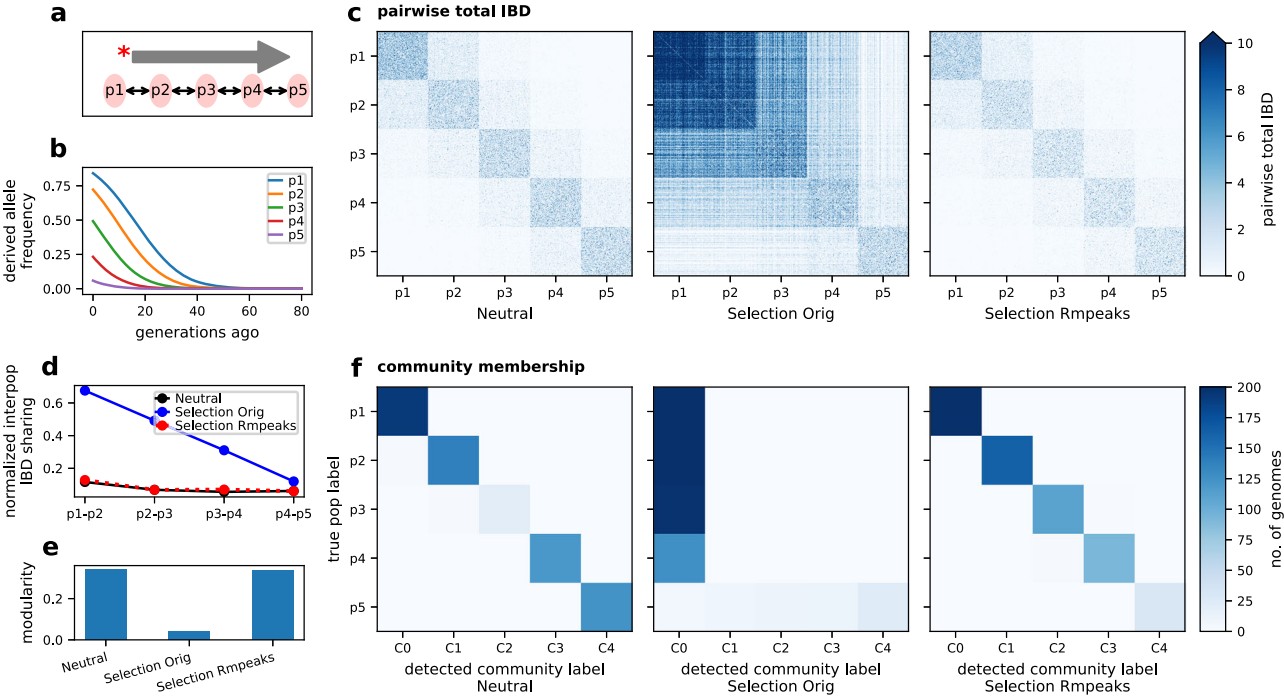

**Fig. 3 | Effects of positive selection on the IBD-based population structure inference. a** Schematic of the one-dimensional stepping-stone model[55] with spreading selective sweeps. Five subpopulations (p1 to p5) were split from an ancestral population. There is symmetrical migration between adjacent subpopulations. A favored allele was introduced into the deme from one side of the chain and spread to the other side. **b** Frequency trajectory of favored alleles (average over chromosomes) in different subpopulations. **c** Heatmap of pairwise genome-wide total IBD under neutral, selection ($s$ = 0.3, Selection Orig), and selection with IBD peaks removed (Selection Rmpeaks). Rows and columns are ordered by true population labels. **d** Normalized inter-population IBD sharing between nearby subpopulations. **e** Modularity of IBD networks with respect to the true population labels before and after removing IBD peaks. **f** IBD network InfoMap community detection before (left, and middle) and after (right) removing IBD peaks. For each subplot, rows are true subpopulations labeled as p1–p5 (assigned in simulation), and columns represent the largest 5 detected communities labeled as C0–C4 (with columns re-ordered to facilitate the comparison of true and inferred labels). The color of each block represents the number of genomes with the given true labels and detected community labels, with darker colors indicating a larger number of genomes. Source data are provided as a Source data file. For results for different selection coefficients and repeated simulations, see Supplementary Fig. 6 and Supplementary Table 1.

distinguishable and consistent with the true population labels (Fig. 3f [right panel]).

## Genome-wide IBD sharing and selection signals in SEA *Pf* isolates

To evaluate the effects of positive selection on IBD-based inferences in empirical *Pf* data sets, we first identified genomic regions that are under positive selection using similar methods as for simulated data, and then compared IBD-based inferences of *Pf* $N_e$ and population structure using IBD segments before and after peak removal.

Previous studies have shown that IBD can be used as a metric to identify genomic regions under positive selection, including regions harboring drug-resistance mutations[9,41]. Based on the same logic, we identified genomic regions with high IBD sharing (see Supplementary Fig. 4 and "Methods") and correlated them with known drug-resistance genes. For empirical data (without genealogical trees generated from simulations), we chose to use the haploid-genome-oriented HMM-based IBD caller (hmmIBD) for IBD inference[60]. The IBD coverage profiles called by hmmIBD show high IBD sharing surrounding: (1) known drug resistance genes and genes associated with the genetic architecture of resistant parasites, such as multidrug resistance protein 1 *pfmdr1*[61], amino acid transporter gene (*pfaat1*)[41,62], chloroquine resistance transporter gene (*pfcrt*)[63,64], dihydropteroate synthase gene (*dhps*)[65], protein phosphatase gene (*pph*, PF3D7_1012700)[66], GTP cyclohydrolase I gene (*gch1*)[67], *kelch13*[68–70], and apicoplast ribosomal protein S10 gene (*arps10*)[66]; (2) genes related to altered sexual investment and increased transmission potential of resistant parasites, including Apicomplexan-specific ApiAP2 family genes *ap2-g* and *ap2-g2*[43,71] (Fig. 4).

When including all parasite genomes, highly related and distantly related, we observe a high level of average (baseline) IBD sharing across the genome in SEA (Fig. 4a) compared to other geographic regions, such as WAF (Supplementary Fig. 7), which is consistent with declines in malaria transmission owing to intensive elimination efforts in SEA[9]. To avoid the confounding effects of high relatedness on further analysis, we applied a heuristic method to remove highly related

isolates by iteratively excluding the isolate with the highest number of strong connections (defined as IBD sharing larger than half of the genome) with others. The resulting subset of isolates, hereafter called unrelated isolates, exhibits a five-times lower baseline IBD proportion (Fig. 4b). The low baseline IBD sharing is less noisy and more readily allows the identification of IBD peaks, including those surrounding *pfmdr1*, and *pfaat1* (Fig. 4b). Thus, we used unrelated isolates and IBD peaks identified from this subset for downstream analyses.

## IBD-based inference of $N_e$ and population structure in a low transmission setting with high background parasite genetic relatedness

Our simulation analyses showed that strong positive selection significantly impacts the IBD distribution, as well as demography and population structure inferences. In these simulations, removing IBD segments within IBD peak regions significantly improved the accuracy of $N_e$ estimates and structure inferences. We assessed this pattern in empirical data from a low transmission setting (SEA).

First, we estimated $N_e$ before removing IBD peaks. Given the high relatedness of parasite isolates in the full data set (Fig. 4a), we focused on the unrelated isolates ($n = 701$). The $N_e$ estimates based on IBD before removing the peaks via IBDNe suggest a decreasing pattern of $N_e$ in SEA, from around $10^4$ to less than $10^3$ in the most recent 60–80 generations (Fig. 5a, blue), consistent with a rapid decrease in malaria incidence in the last decades owing to malaria elimination efforts in this geographic region[1].

Second, we performed population structure inference before peak region removal via IBD network analyses. Using the total IBD matrix for unrelated isolates as input, we performed unsupervised community detection via InfoMap. Among the 701 unrelated isolates from SEA, we identified five communities (defined via IBD network structure analysis instead of geopolitical boundaries) with sizes > 20 isolates. The largest community, labeled as C0 (Fig. 5b) was enriched for parasites sampled from Western Cambodia (Battambang, Kratie, Oddar Meanchey, and Pursat) (Supplementary Fig. 8 and

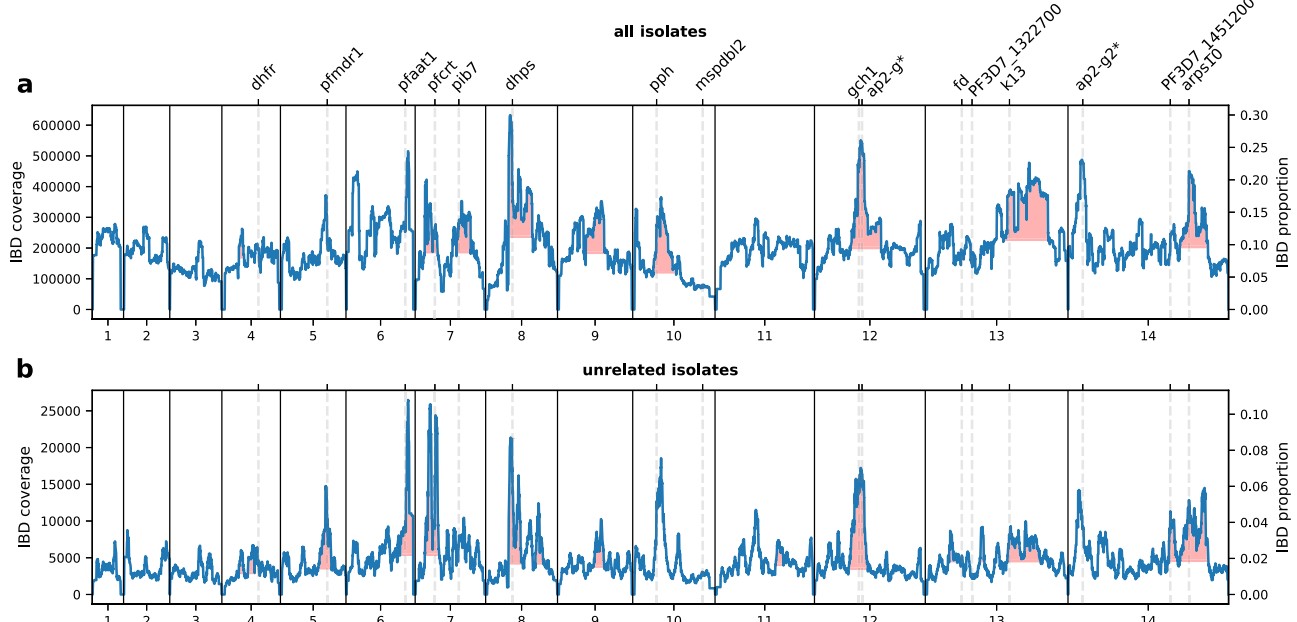

**Fig. 4 | IBD coverage profile of all and unrelated *Pf* isolates in SEA. a** IBD coverage/proportions of all parasite genomes ($n = 2055$), including highly related and unrelated, in SEA. Labels on the top indicate the center of known or putative drug-resistance genes or genes involved in sexual commitment (*). **b** IBD coverage/proportions of unrelated genomes in SEA ($n = 701$). Annotations in (**a**) are shared with (**b**); regions with red shading indicate validated peaks (defined in "Methods").

Note: (1) different scales for y axes (IBD coverage on the left y-axis; IBD proportions on the right y-axis) were used in (**a**) versus (**b**) to better reveal the peaks; (2) the peaks around *pph* in (**b**) and *ap2-g* in (**a**) and (**b**) are IBD peak candidates that do not pass the peak validation step (see "Methods"). Source data are provided as a Source data file.

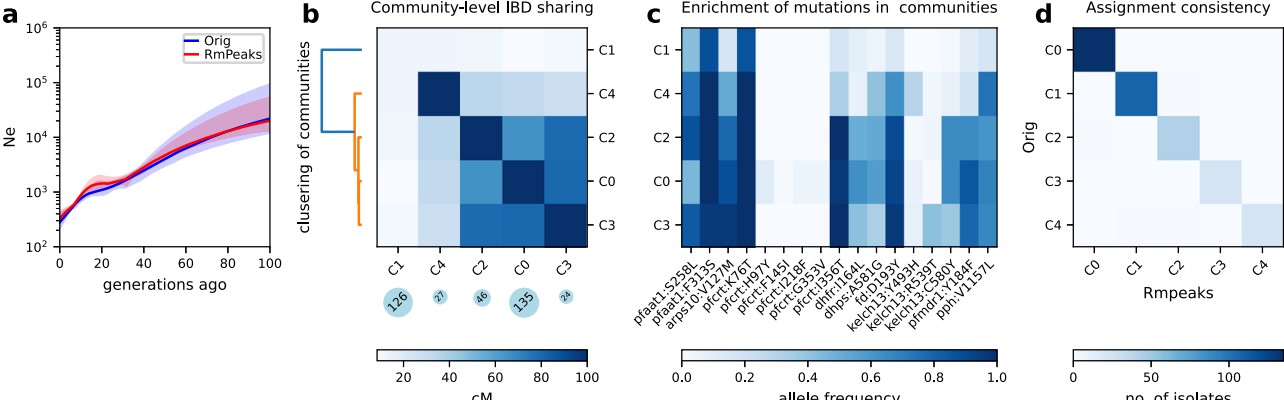

**Fig. 5 | $N_e$ and population structure inference in an empirical data set from SEA.** **a** $N_e$ estimates for SEA before and after removing IBD peaks. Error bands indicate 95% confidence intervals as determined by IBDNe[12]. **b** IBD network analysis of SEA data (before removing IBD peaks), including community-level IBD sharing matrix (heatmap), community size (blue circles below), and dendrogram showing hierarchical clustering of the community-level IBD matrix (left). Only the largest 5 communities, labeled as C0 to C4, are plotted. The rows and columns in the heatmap, each representing one of the 5 communities, are re-ordered such that heatmap and hierarchical clustering share the detected community labels (y axis tick labels). **c** Frequency of drug resistance mutations in different IBD communities. **d** Consistency of InfoMap assignment of unrelated isolates before (x-axis) and after (y-axis) removing the peaks. Source data are provided as a Source data file.

Supplementary Fig. 9a). In contrast, the second-largest community (C1) was comprised of isolates from a wider geographic area, including Northeastern Cambodia (Ratanakiri), Laos, and Vietnam (Supplementary Fig. 8 and Supplementary Fig. 9a). Isolates within C1 were distantly related given the low-average within-community IBD sharing (Fig. 5b, top left block) compared with other communities. Hierarchical clustering of the community-level average IBD sharing matrix revealed other major communities such as C2/3/4 are closer to C0 rather than C1. The detected communities also displayed temporal dynamics. For instance, parasites from Pursat, initially belonging to different communities, converged into the main community C0 in recent years; a similar pattern occurred in Oddar Meanchey 2–3 years later relative to Pursat (Supplementary Fig. 9b, c). These changes are consistent with the spread of artemisinin-resistant parasite lineages from the west to the north over time.

To investigate the potential drivers of the observed population structure, we examined the distribution of non-synonymous mutations in known drug-resistance genes in the detected communities. We discovered that different communities or community groups exhibit distinct mutational landscapes at drug resistance loci (Fig. 5c). The group clustered with C0−C0/2/3/4−demonstrates relatively high frequencies of several resistance variants, including those associated with artemisinin-resistance or its associated genetic background, e.g., *arps10* V127M, *pfcrt* I356T, ferredoxin (*fd*) D193Y, *kelch13* C493H, R539T, and C580Y, and *pph* V1157L, as well as mutations associated with resistance to other antimalarial drugs, e.g., *dhfr* I164L and *dhps* A581G (associated with resistance to the antifolates) and *pfcrt* H97Y, I218F and G353V (mutations associated with piperaquine resistance found in C0)[8,66,72]. In contrast, the C1 communities had relatively low frequency or no mutations (including *kelch13* mutations) at these loci. The mutation landscapes for the largest 5 communities show distinct *kelch13* resistance mutation patterns, consistent with the presence of multiple artemisinin-resistant founder populations and an artemisinin susceptible population previously observed in this geographic region[8]. These results suggest that the population structure of *Pf* in SEA is heavily influenced by drug resistance and positive selection and confirms that different founder populations harbor distinct combinations of resistance mutations[8].

Finally, we compared the IBD-based inferences before and after peak removal. In the SEA data set, the removal of IBD peak regions did not significantly alter estimates of $N_e$. The trajectories of $N_e$ estimates heavily coincide, with point estimates of pre- (Fig. 5a, blue) and post-IBD-peak removal (Fig. 5a, red) having overlapping 95% confidence intervals. The inferred population structure patterns were also similar before and after the correction, with the community assignments before removing IBD peaks being consistent with those after IBD peaks were removed (Fig. 5d). Although there are some minor changes in population structure inference, the size of the main communities and the Adjusted Rand Scores[73] are largely unchanged (Supplementary Table 2).

## Effects of removing IBD peaks on IBD-based inferences in a high transmission setting with low background parasite genetic relatedness

The effects of positive selection on IBD-based inferences observed in simulations are not corroborated by the empirical results observed in the analysis of data from parasites sampled in SEA. We hypothesized that this discrepancy stemmed from high baseline genetic relatedness observed in recent time frames in SEA (Supplementary Fig. 10), even after having pruned highly related isolates (the equivalent of first-degree relatives) (Fig. 4b versus Supplementary Fig. 7b). To test this hypothesis, we took two approaches: (1) we incorporated high relatedness/inbreeding into simulations; and (2) we evaluated an empirical data set from a high transmission setting, WAF, where parasite relatedness and inbreeding are known to be lower than in SEA[9].

We modeled high relatedness/inbreeding in our simulations in three different ways, by incorporating decreasing population size (Supplementary Figs. 11 and 12), positive assortative mating (Supplementary Figs. 13 and 14) and selfing (Supplementary Figs. 15 and 16), to simulate parasite populations with different levels of inbreeding. Our results show that, in high-inbreeding populations simulated via decreasing population size or positive assortative mating (Supplementary Figs. 11–14, bottom rows), selective sweeps have a lower local impact on the genome (measured as peak impact index) and a higher, chromosome-wide global impact (measured as global impact index, Supplementary Fig. 17 row 3; see Supplementary Note 1 for definitions). Thus, removing IBD peaks, in this case, provides a less effective and necessary bias correction (estimates are similar before and after IBD peak removal) for both $N_e$ estimation (Supplementary Figs. 11 and 13) and population structure inference (Supplementary Figs. 12 and 14), when compared with low-inbreeding simulations (Supplementary Figs. 11–14 top rows). In these low-inbreeding simulations, inbreeding level tends to be negatively correlated with local impact on the genome and positively correlated with global impact on the genome (Supplementary Fig. 17, left two columns). The pattern is different when inbreeding is modeled via selfing, where both local

impact and global impact increase with the inbreeding levels (Supplementary Figs. 15 and 16, and Supplementary Fig. 17 last column), likely due to a synergistic effect of selfing on selection (i.e., a faster increase in the frequency of selected alleles in the presence of a high selfing rate, Supplementary Fig. 15c). Despite these differences, the results suggest that selective sweeps tend to have strong global impact in populations with high levels of inbreeding and low effective recombination rates; in this case, IBD-peak removal-based bias correction is either less necessary or less beneficial (see Supplementary Note 2 and 3 for detailed methods and analyses).

Second, to further test our hypothesis, we explored the effects of removing IBD peaks in a high malaria transmission setting by analyzing parasite isolates from WAF. Using the same criterion for unrelated isolates as in the SEA data set, we found that removing IBD peaks in the WAF data set ($n = 1496$ unrelated isolates), resulted in larger estimates of $N_e$ for the most recent generations, with non-overlapping confidence intervals approximately 20 generations ago (Fig. 6a). Additionally, we were able to uncover finer population structure using IBD network-based community detection after removing IBD peaks. Before selection correction, most isolates were collapsed into a dominant community (Fig. 6b), while after IBD peak removal, these isolates were assigned to multiple smaller communities (Fig. 6c), with differences in the inferred population structure before and after IBD peak removal being statistically significant based on Jackknife resampling (Supplementary Table 2). The change in detected communities before and after removal of IBD peaks is similar to the results from simulations without high-relatedness or inbreeding (Fig. 3). Together, these results suggest that background genetic relatedness and inbreeding are key modifiers of the effect of selection on IBD-based inferences and the necessity for bias correction.

## Discussion

*Pf* parasites have undergone strong and multiple selective sweeps owing to selection for resistance to antimalarial drugs, which is known to alter IBD patterns. Our simulations and analysis of field isolates demonstrated that strong positive selection can alter IBD distributions, resulting in an underestimation of $N_e$ and a blurring of population structure. Our new IBD peak removal strategy can partially mitigate this bias, particularly in areas with low background relatedness (high transmission settings). Thus, we recommend excluding genomic regions under positive selection when using IBD-based approaches to estimate parasite population demography in areas

with high malaria transmission rates and low parasite genetic relatedness, such as sub-Saharan Africa. However, in scenarios with low transmission and high relatedness, as in SEA, correction for selection may not be necessary, as such correction did not significantly change IBD-based inferences.

Although positive selection is known to increase the likelihood of allele IBD[9,37,40] and accumulation of longer haplotype blocks surrounding the selected sites[39,74], the effect of recent positive selection on the IBD segment distribution in the *Pf* genome is difficult to assess owing to the action of opposing and confounding factors, such as: (1) strong selection that increases local IBD sharing in the genome; (2) high genome-wide background IBD sharing (due to decreasing $N_e$ and structuring of parasite population) that hides selection effects; and (3) low SNP density (due to high recombination) that affects IBD call quality and the evaluation of selection effects. Our simulation approach circumvented IBD quality issues by using true IBD and mimicked high background IBD sharing by introducing high relatedness and inbreeding into the simulations. We provide evidence that positive selection can significantly alter IBD distributions despite the above complex context of *Pf*. Thus, ignoring selection bias in *Pf* could lead to inaccurate IBD-based inferences, particularly in high transmission settings.

The change in IBD segment length due to selection can have significant implications for time-specific inferences. This shift challenges the assumed correlation between the length of an IBD segment and its age (TMRCA), as IBD segments tend to be longer in selected regions than in neutral regions even from ancestors of similar ages. As a result, IBD-based $N_e$ estimators (such as IBDNe[12]), time-specific migration surface estimators (such as MAPS[22]), and other time-specific IBD analyses that rely on TMRCA estimates inferred from IBD segment length[24,53], could all suffer from selection bias. Similarly, LD-based $N_e$ estimates[75] are potentially biased by strong positive selection because of increased long-distance LD, which can violate the assumption that LD over longer genetic distances is informative for $N_e$ estimation in more recent time frames and over shorter distances for $N_e$ at distant time frames[76,77]. Although our finding is consistent with the predicted influence of selection on the estimates of $N_e$[77], it contrasts with the recent finding that LD-based $N_e$ estimates are virtually unaffected by natural selection[78]. The different conclusions are likely due to the distinct ways of simulating selective sweeps. Novo et al. used smaller selection coefficients and did not strictly control selection loci and time (randomly chosen), which generally focuses on the relatively

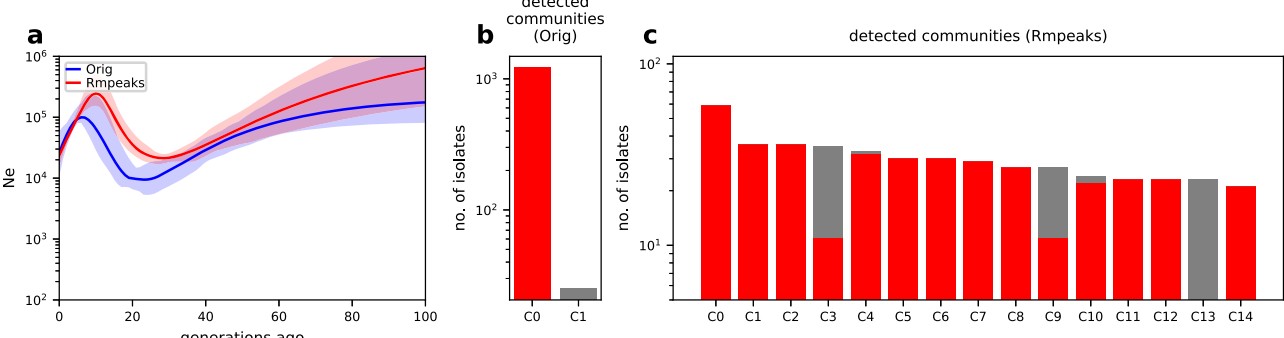

**Fig. 6 | Removing IBD peaks changes the inference of $N_e$ and population structure in the West African (WAF) data set. a** $N_e$ estimates of *Pf* in the WAF data set before (blue) and after (red, dotted) removing IBD peaks. Error bands indicate 95% confidence intervals as determined by IBDNe[12]. **b, c** Distribution of the sizes of detected communities from IBD networks using the WAF data set before (**b**) and after (**c**) removing IBD peaks. Only communities with at least 20 isolates are shown. The y axis indicates the number of isolates assigned to a detected community. The x-axis tick labels are the detected communities labeled as C0, ..., C(n−1). Note that in (**b**) the leftmost red bar labeled C0 represents the dominant community in the

original IBD network, with a size of 1222 isolates. Panel (**c**), on the other hand, shows the distribution after removing IBD peaks. This process leads to a reassignment of isolates from the dominant community (C0 in (**b**)) into smaller, distinct communities, labeled as C0-C14 in (**c**). To visually convey how community assignments have shifted as a result of this reassignment, each bar in (**c**) is split into two color components: red and gray. The red portion represents isolates that were part of the original dominant community (community C0 in (**b**)), while the gray portion indicates isolates that are not from this dominant community. Source data are provided as a Source data file.

long-term effects of less-constrained, weak selective sweeps that mimic the human scenario. We instead conditioned on the establishment of positive selection at specific loci starting at designated time points such as 50 generations ago which mimics the recent, strong, and multi-locus selective sweeps observed in empirical *Pf* data.

The effect of selection on the distribution of IBD segment length can similarly impact estimates of genome-wide IBD, such as pairwise genome-wide total IBD sharing. Pairwise total IBD, similar to the fraction of sites sharing IBD, is a measure of genetic relatedness used for network-based analysis of population structure[6,9,14,17,25,53,79,80]. As an aggregate of IBD shared across the genome, including that from neutral and non-neutral regions, pairwise total IBD-based relatedness estimates can be driven by at least two components, genomically local enrichment of IBD (due to non-neutral evolution), and genome-wide alteration in IBD sharing (due to changes in population structure, $N_e$, and inbreeding). Positive selection increases pairwise total IBD by increasing IBD sharing around selected loci (chromosomally local). In a panmictic malaria parasite population of large $N_e$ and low inbreeding where average IBD sharing is low (such as in Africa), pairwise total IBD is dominated by the excess local IBD due to selection, which yields artificial patterns of relatedness unrelated to subpopulation structure. Thus, failure to correct selection bias can cause overestimation of genetic relatedness and make underlying fine-scale population structure (defined by neutral regions) hard to discern.

Given the presence of selection bias, it is important to identify and correct the bias to improve the accuracy of IBD-based analyses. While there are existing heuristic methods to classify genomic loci as selected, linked neutral, or neutral[81,82], there are no specific methods to filter IBD segments for correction of positive selection-induced biases. One related exception is the IBDNe program[12], which internally excludes regions sharing extremely high IBD. Our peak removal strategy borrows from this idea but lowers the peak identification threshold for selected loci, expanding the selected region to the chromosomal median, in an attempt to include neutral loci that are linked with the selected locus. Our work demonstrates that this peak removal method can successfully mitigate selection-induced bias in IBD-based inference of $N_e$ and population structure.

The influence of positive selection on IBD-based demographic inferences is closely linked to the baseline level of relatedness or IBD sharing across the genome. In malaria parasite populations characterized by high average IBD sharing (indicating high background genetic relatedness), there tends to be increased inbreeding and reduced effective recombination[83]. In such populations, strong positive selection can have expansive effects, potentially impacting broad genomic regions (as shown in Supplementary Figs. 15 and 16, peak width) or even entire chromosomes[84] (as depicted in Supplementary Fig. 17, based on our global impact index). The magnitude of selection bias and the need for peak-removal-based correction are influenced by both the local and global impacts of these selective sweeps on the genome. Generally, given an identical selection coefficient, selective sweeps occurring in populations with higher inbreeding exhibit a more pronounced global impact on the genome, which inversely correlates with the effectiveness of and necessity for peak-removal-based correction. Conversely, in populations with lower background genetic relatedness where selective sweeps have a limited global impact, the local impact (measured by our peak impact index) can serve as a valuable quantitative metric to determine the need for selection correction (see Supplementary Note 1 for instructions on how to calculate peak impact index). For example, based on our analyses we might recommend removal of IBD peaks with a peak impact index > 0.01; however, sensitivity analyses using different thresholds is recommended. We caution against selection correction in scenarios with high background relatedness. In these cases, correction is often less effective at enhancing the accuracy of IBD-based estimates and may lead to over-correction. This over-correction could arise from a secondary shift in IBD distribution, where cutting IBD segments at peak region boundaries inadvertently increases the frequency of shorter IBD segments.

Besides selection bias, the application of IBD-based approaches in *Pf* research can potentially suffer from IBD call quality issues, for instance, due to relatively low SNP density (low ratio of mutation to recombination rate). Currently, IBD calling for *Pf* is performed predominantly via one of two HMM-based tools, isoRelate or hmmIBD, which report the fraction of site IBD (allele IBD) as the metric of genetic relatedness, with the individual IBD segments not fully evaluated or utilized[9,60]. Our simulation models and true IBD algorithm designed for selection bias evaluation provide the foundation for development of an IBD benchmarking framework for high recombining species (ongoing work). This benchmarking framework will differ from the methods used by the top two IBD callers for *Pf*. While hmmIBD and isoRelate were validated via simple pedigree-based simulations and parent-offspring trio simulations, our framework utilizes genealogy simulation and recording (msprime/SLiM, tskit respectively)[51,52,85], and a tree sequence-based IBD inference algorithm (inspired by ref. 12) to provide a complementary, but more flexible population-based framework for benchmarking IBD quality for different IBD callers, or evaluating IBD-based methods in non-human species. The importance of this IBD-validation framework is also highlighted in an independent work focusing on benchmarking IBD callers for human genomes[86].

Despite our efforts to conduct comprehensive analyses, our present work is accompanied by several caveats. First, *Pf* evolutionary parameters are not all well-characterized and vary greatly across studies. Thus, we had to make assumptions about the realistic values of these parameters and ignore genome heterogeneity for simulation. These assumptions, if biased, might affect the accuracy of simulation analyses. Second, we simulated single-site selection simultaneously on more than one chromosome to mimic multiple selective sweeps, but this approach may inflate the positive selection-induced bias of IBD-based estimates. In real-world scenarios, selective sweeps may occur sequentially as drug policies change over time. Third, the empirical data has high heterogeneity in sampling location and time. The mixture of isolates from different years might complicate temporal interpretation of $N_e$ estimates over past generations and the correspondence of IBD segment length with TMRCA. Furthermore, the presence of structure among isolates from different geographic regions as reported previously in SEA could violate the population homogeneity assumption of IBDNe[12] and bias $N_e$ estimates. This could be partially addressed by running $N_e$ estimation on isolates from a smaller time window and a specific geographic region if high variation due to a smaller sample size is tolerable. The small sample size issue could be resolved by incorporating data from the recently released, much larger MalariaGEN *Pf* 7 WGS database[87].

In conclusion, our study demonstrates the impact of strong positive selection on IBD-based estimates of $N_e$ and population structure in *Pf*. We show that removing excess IBD within genomic regions corresponding to selective sweeps can partially correct the biases induced by positive selection and emphasize the importance of considering selection effects when using IBD-based methods for either demography inference or population structure analysis, particularly in high malaria transmission areas. Our novel population simulation and true IBD inference framework, which employs flexible selection simulation and tree-sequenced-based IBD inference, provides a valuable tool for benchmarking IBD callers and evaluating IBD-based methods in species with extreme evolutionary parameters relative to humans.

## Methods

### Parasite isolates and whole-genome sequencing
The analyzed data were obtained from a publicly available repository (MalariaGEN Catalogue of Genetic Variation in *P. falciparum* v6.0, *Pf*6)

 

or from our in-house sequencing data sets (NCBI SRA under access number PRJNA1004408 and PRJNA312679). Information about informed consent for the publicly available MalariaGEN data have been described previously[48]. In-house *P. falciparum* WGS data were generated from samples collected from individuals with symptomatic malaria participating in research studies conducted by the Armed Forces Research Institute of Medical Sciences with approval from the Walter Reed Army Institute of Research and local ethics committees. All study participants or their guardians provided informed consent. Parasite sequencing and genomic analyses were undertaken with the approval of the University of Maryland School of Medicine Institutional Review Board. Patient clinical and demographic information (e.g., sex and age) were not used in the analyses, which focused strictly on parasite population genomics. DNA was extracted from leukocyte-depleted blood samples using a Qiagen DNA Midi Kit (Qiagen, Hilden, Germany) and sequencing libraries generated using the KAPA Library Preparation Kit (Kapa Biosystems, Woburn, MA). Whole-genome sequencing was performed on an Illumina HiSeq 4000 or Illumina Novaseq 6000 (Illumina, San Diego, CA) using 150 bp paired-end reads.

We performed joint variant calling on the WGS data using a unified variant calling pipeline (see snp_nf_call pipeline in "Code availability") that follows GATK best practices and the MalariaGEN *Pf*6 data-generating protocol[48]. Briefly, raw reads were first mapped to the human GRCh38 reference genome to remove host reads, with the remaining reads being mapped to the *Pf*3D7 reference genome (PlasmoDB_44). Mapped reads were processed using GATK (version 4.2.2) MarkDuplicates and BaseRecalibrator tools, after which analysis-ready mapped reads for each isolate were used to generate per-sample calls (HaplotypeCaller/GVCF mode). These per-sample calls were combined and run through a joint-call step (GenotypeGVCFs) to obtain unfiltered multi-sample VCFs. We then used a machine learning-based variant filtration strategy, GATK VariantRecalibrator, to retain high-quality variants. Only biallelic SNPs were used in our analyses. Sites and samples were filtered based on genotype missingness and allele frequency, ensuring both per-sample and per-site missingness for filtered data were less than 0.3 and minor allele frequency ≥0.01.

Polyclonal parasite isolates were identified by calculating $F_{ws}$, a metric analogous to Wright's inbreeding coefficient, that is used to distinguish monoclonal from complex infections[88]. Monoclonal isolates were defined as those with $F_{ws} > 0.95$ (inferred by the moimix package, version: 802eaf1)[89], and phased using dEploid (version: v0.6-beta)[90], with missing genotypes imputed using Beagle (version: 5.1)[91]. Monoclonal isolates then served as the reference panel to extract and phase the predominant clones in polyclonal isolates, using dEploidIBD (version: v0.6-beta)[92], of which the performance in predominant haplotype deconvolution has been verified by single-cell sequencing[79]. To further ensure high deconvolution quality, we used the inferred ratios of clones (haploid genomes) within an isolate to determine whether a polyclonal infection has a dominant clone: the ratio of the major clone should be greater than 0.7 and at least 3 times larger than the minor clone. The combined biallelic, phased, and imputed data, including haploid genomes from the monoclonal infections and dominant clones from the polyclonal infections, were used in downstream analyses. The details of empirical data analyses are available in the posseleff_empirical pipeline (see "Code availability").

### Single-population genetic data simulation

In general, simulated data were generated using forward (SLiM, version: 4.0.1)[51] and coalescent (msprime, version: 1.2.0)[85] simulators and encoded in tree sequence format[52,93]. The advantages of combining these simulators include their efficiency (msprime), and flexibility in simulating different modes of selection (SLiM), demography, and structure. We used SLiM to perform forward-time simulations with selective sweeps of different parameter values. We then performed

simplification and coalescent simulation steps[52]. Briefly, as SLiM explicitly models each diploid individual, the recorded tree sequences from SLiM simulation contain all explicitly modeled individuals (equal to population size N), which is more than the specified sample size. We simplify the tree sequence by subsetting so that only the requested number (sample size) of present-day genomes are kept. As the simplified tree sequences might not have fully coalesced (for instance, having multiple roots), we ran an additional backward-time simulation via msprime to fill in the top of genealogical trees, and ensure that trees coalesce into a single root, the grand common ancestor. These steps, including the forward simulation, simplification, and coalescent simulation, together generate the full genealogy ancestry for the specified samples. The full ancestry (without mutation information) was then used for two purposes: (1) to generate true IBD segments, which are elaborated below and only rely on tree topology, and (2) to allow the addition of simulated mutations via msprime onto geological tree branches and the generation of phased genotype data in variant calling format (VCF). The simulated genome consists of 14 chromosomes, each with a length of 100 centimorgans (cM), the total of which resembles a real *Pf* genome. We assume a constant recombination rate of $6.67 \times 10^{-7}$ per generation per bp (15 kb/cM)[41,44]. The mutation rate was assumed to be $1 \times 10^{-8}$ per generation per bp[28,84] based on which phased genotype data (VCF file) is generated. Of note, mutation information is not required for tree topology-based true IBD generation.

To evaluate the effect of selection on IBD distribution and IBD-based $N_e$ inference, we simulated data using the single-population model, where no population structure and migration are allowed. Under this model, we simulated genealogies with varying values of selection parameters, including selection strength, number of origins, and selection starting time. Due to genetic drift (especially for low initial allele frequency or weak selection coefficient), the favored allele (under selection) can be lost and in turn no effective selection would be observed in the present-day samples. To condition on the establishment of selection, we rerun the simulation with different seeds up to 100 times until the favored allele is not lost at the present-day time.

We ran simulations under different $N_e$ scenarios, including constant $N_e$, and exponential decrease, where the ancestral population size was assumed to be 10,000. Our choice of $N_e$ 10,000 is an intermediate value within the large range of $N_e$ estimates for *Pf* in Southeast Asia from $10^3$ to $10^5$ [27,94,95]. To evaluate the effect of selection on IBD distribution and $N_e$ estimates, we chose to use the exponential decrease demographic model to mimic the *Pf* demography in SEA.

The details of simulation studies, including single- or multiple-population simulations, inbreeding modeling, and associated downstream analysis can be found in posseleff_simulations pipeline (see "Code availability").

### IBD calling and removal of highly related isolates

We implemented a true IBD inference algorithm (tskibd, version: v0.0.1, available at: https://github.com/bguo068/tskibd) based on true genealogical trees, to circumvent the bias due to low-quality IBD calls from mutation information (phased genotype data), and directly test the effect of selection on IBD and IBD-based inferences. The tskibd algorithm was implemented on top of the tskit C API[85]. Our definition of IBD segments closely follows Browning et al.'s work[20]. The main concept in this algorithm is that for each pair of haploid genomes, if the most recent common ancestor (MRCA) stays the same across adjacent marginal trees so that the span is longer than a specified threshold such as ≥2 cM, this unbroken, long, and shared ancestral segment is defined as an IBD segment for this pair of haploid genomes. The algorithm produces IBD segment records for all pairs of haploid genomes, chromosome by chromosome. In addition to start and end coordinates, the produced IBD records contain additional useful information, such as whether the segment contains the favored allele

(if recorded from SLiM simulation) and MRCA and its age, on which we can depend for accurate segment filtration that could not be done using other IBD callers. We apply this algorithm for all simulated data where true genealogy is already available.

When working with an empirical data set, the true genealogy is not available. We chose to use hmmIBD (version: a2f796e) for IBD calling from our empirical data set as it was developed specifically for haploid parasites including *Pf* and has been commonly used in the malaria research community[60,96,97]. As instructed in hmmIBD documentation, we modified the recombination rate in the source code to make it consistent with our simulations (15 kb/cM) and specified the -n option to 100 to call IBD segments from MRCAs in the recent 100 generations. The input includes phased and imputed genotype data from all monoclonal samples and the dominant haploid genome from polyclonal samples (see criteria above). Tracts < 2 cM in length were excluded, as these short tracts lack statistical support to be confidently called IBD[9,20].

We used a heuristic method to remove isolates that are highly related while maximizing the number of remaining isolates. First, we defined a pair of isolates (represented by haploid genomes, either from monoclonal infections or the dominant clone from polyclonal infections) as highly related if their genome-wide (chromosome PF3D7_01_v3 to PF3D7_14_v3 for empirical *Pf* data) IBD sharing is no less than half of the genome size. We then built an adjacency matrix including all highly related isolates. The relatedness of each pair is either high (represented by 1) or low (represented by 0). We made a network/graph based on the above high relatedness adjacency matrix. From the graph, we iteratively deleted the node with the highest number of connections (degrees) and the node's related connections and append the deleted node to a list until no connection was present in the graph. The list of deleted nodes (from the high relatedness network) is the subset of isolates to be removed such that all remaining isolates are unrelated (not highly related). Finally, we removed all IBD segments involving the to-be-deleted isolates from input IBD. IBD shared among unrelated isolates was used for $N_e$ and population structure inference.

### IBD coverage profiling and peak identification and removal

We calculated IBD coverage as the number of IBD segments overlapping each position of a list of evenly spaced (0.1 cM) sampling points along each chromosome. To allow comparing IBD coverage across data sets with different sample sizes, we calculated IBD proportion for each sampling point as its IBD coverage divided by the total number of possible pairs of haploid genomes in a data set. IBD peak candidates were identified as regions with IBD sharing higher than two 5%-trimmed standard deviations above the trimmed mean (core region) which is extended on both sides until the coverage reaches the median coverage of the chromosome (extension regions). To differentiate noise from real selection signals, we calculated an integrated haplotype score[54] (iHS)-based positive selection statistic, $X_{iHS}$ for each SNP with minor allele frequency > 0.05. Unstandardized iHS scores calculated via scikit-allel (version: 1.3.5) ihs function are standardized within their allele frequency bins (standardize_by_allele_count)[98]. We squared the standardized iHS score so that the new summary statistic, $X_{iHS}$, follows a chi-squared distribution with 1 degree of freedom. We treated each SNP with a statistically significant $p$ value as a hit. We then overlaid IBD peak candidates and $X_{iHS}$-based hits. An IBD peak candidate that contains ≥1 hit was defined as a verified IBD peak.

The identified and verified peak regions (core and extension regions) correspond to selective sweeps. We evaluated the local impact of each peak via the peak impact index (defined in Supplementary Note 1) and marked peaks with an impact index > 0.01 as target peaks for removal. For each IBD segment, we removed the whole segment if it was contained within a target peak region, or removed the part of the segment that was overlapping with any target

peak regions. The remaining segment(s) longer than 2 cM were retained. The removal of IBD peaks creates empty ranges of zero IBD coverage. When preparing input for IBDNe[12], we followed the internal algorithm used in IBDNe and split chromosomes into contigs by treating each contiguous region with non-zero IBD coverage as an independent contig (chromosome). We implemented the peak identification and removal algorithm in the ibdutils Python package (see "Code availability").

### Effective population size estimation

The trajectories of $N_e$ were estimated via IBDNe (verion: 23Apr20.ae9). We used IBD segments no shorter than 2 cM as input for IBDNe. To be compatible with tools designed for diploid species, such as IBDNe, we processed the haploid-level IBD to diploid-level IBD by treating each haploid genome (isolate) as a pseudo-homozygous diploid. Processed IBD segments and a constant-rate (15 kb/cM) recombination map were used to estimate $N_e$ over the last 150 generations using IBDNe, with a focused interpretation on the latest 100 generations[12]. The minregion parameter was changed from 50 to 10 cM to include shorter *Pf* contigs/chromosomes. For selection-aware $N_e$ estimation, we removed and/or split IBD segments corresponding to IBD peaks as described above. The default bootstrap sampling parameter values (nboots = 80) was used to estimate uncertainty for empirical data sets. The final $N_e$ estimate is scaled by a factor of 0.25, to account for the haploid to homozygous diploid conversion.

### Multiple-population genetic data simulation

To evaluate the effects of selection on population structure, we simulated genealogies of genomes under positive selection involving multiple subpopulations using a 1-dimensional stepping-stone model. Five subpopulations, split from the same ancestral population ($N_e$ = 10,000) 500 generations ago, each with an effective population size of 10,000, are connected by symmetrical migration between neighboring populations (p1 ↔ p2 ↔ p3 ↔ p4 ↔ p5, with a migration rate of $10^{-5}$ until selection starts). A favored allele (hard selective sweep) originates from the leftmost population (p1) at a specified time (such as 80 generations before the present), expands and spreads toward the rightmost population (p5) via migration. The selection pressure (with selection coefficients varying from 0 to 0.3) on this allele is the same across all populations (i.e., uniform selection rather than heterogeneous selection). With appropriate combinations of the selection and migration parameters (0.01 is used after selection starts), this model mimics the gradient of allele frequencies that occur when a selective sweep spreads across populations[3]. Similar to the single-population simulation model, we reran each simulation up to 100 times until the present-day allele frequency in the subpopulation p2 is at least 0.5, in order to reduce the possibility of the favored allele being lost during the sweep process, and measure effects of established positive selection.

### Population structure inference

We inferred the population structure based on a pairwise genome-wide total IBD matrix. For the empirical data set, we used only longer segments (≥4 cM) to build a total IBD matrix that reflects a more recent structure. We set elements of the total IBD matrix to zeros if they are less than 5 cM to reduce noise and reduce the density of the matrix. To test if there is an isolation-by-distance pattern, we plotted a heatmap of an unclustered IBD matrix that was ordered by sampling location or true population label. For unsupervised clustering, we combined two different approaches: (1) Run InfoMap algorithm (a community-detection method implemented in python-igraph[58] package (version: 0.10.0) on an IBD network that is built upon the square-transformed IBD matrix ($y_{ij} = x_{ij}^2$). This square transformation can help better reveal the fine-scale structure within the simulated population. (2) Run an agglomerative hierarchical clustering via the mean linkage criterion on

a community-level average IBD sharing matrix to assess how communities were related. For hierarchical clustering, we converted the IBD matrix from a similarity matrix to a dissimilarity matrix via the formula $\mathbf{Y} = \max(\mathbf{X})/(\mathbf{X} - \min(\mathbf{X}) + \delta)$, where $\delta$ is a small number 0.001.

Additionally, when the true population structure was known (in simulated data from the multiple-population model), we calculated the (1) modularity of the IBD network with respect to the true structure to measure how the genomes within each true population are separated from other groups using the Graph.modularity method in the python-igraph package, and (2) the normalized inter-population IBD sharing defined as the ratio of inter-population sharing $I_{i,j}$ to the square root of the product of the intra-population sharing $I_i$ and $I_j$. To evaluate the effect of removing IBD peaks (selection effects), the above inference or calculation was repeated on IBD data with peak regions excluded.

### Simulation of high background relatedness/inbreeding

Incorporation of high relatedness/inbreeding into the single-population or multiple-population models was implemented in the SLiM simulation script via three different strategies, including incorporation of (1) decreasing population size, (2) assortative mating, and (3) selfing. Simulations with varying values of inbreeding-related parameters were performed to mimic populations with different inbreeding coefficients. The relationship between the level of inbreeding and the impact of selective sweeps was analyzed via three metrics: inbreeding potential, peak impact index, and global impact index. The details of the inbreeding simulations are provided in Supplementary Notes 1–3.

### Statistical methods

To verify IBD peaks, we calculated the IBD-based statistic, $X_{iHS}$, for each SNP as described above. To compare $N_e$ estimates based on filtered (corrected) and unfiltered (uncorrected) IBD inputs for IBDNe, 95% confidence intervals were generated from bootstrap sampling. For simulated data, we used a two-sided Wilcoxon signed-rank test to examine point estimates of 30 replicated simulations before and after removing IBD peaks for each generation. The $p$ values were adjusted using a Bonferroni correction as estimates from nearby generations are highly correlated.

For comparing community-detection-based population assignments in empirical data sets, we calculated the Adjusted Rand Score[73] to quantify the level of consistency between memberships before and after removing IBD peaks. We used jackknife resampling to obtain confidence intervals by randomly excluding one chromosome and rerunning the community detection analyses for each sampling. Assignment differences were considered statistically significant when the upper bound of the confidence interval (the third quartile + 1.5 interquartile range) did not contain the value 1.0, given an expected value of 1.0 for identical assignments. For simulated data with known true population labels, we calculated the Adjusted Rand Scores against the true population labels. To determine uncertainty in the simulations, we ran 30 replicates of the same model and parameter values using different seeds. We compared the Adjusted Rand Scores (each against the truth) before and after removing IBD peaks using a two-sided paired $t$-test. A $p$ value, or an adjusted $p$ value if corrected, <0.05 was considered statistically significant, unless otherwise indicated.

### Reporting summary

Further information on research design is available in the Nature Portfolio Reporting Summary linked to this article.

### Data availability

Reads of new whole-genome sequence data ($n$ = 640 $Pf$ isolates) are deposited to NCBI Sequence Read Archive (SRA) and publicly available under the accession number PRJNA1004408. Other publicly available WGS data can be found in MalariaGEN Catalogue of Genetic Variation in *P. falciparum* v6.0, *Pf*6 (meta information: https://www.malariagen.net/resource/26/; raw reads: https://www.ebi.ac.uk/ena/browser/home, $n$ = 2978 analyzed) and NCBI SRA under the accession number PRJNA312679 (https://www.ncbi.nlm.nih.gov/bioproject/?term=PRJNA312679, $n$ = 111 analyzed)[99]. The accession numbers and hyperlinks for each individual isolate are provided in Supplementary Data 1. Source data for relevant Figures, Supplementary Figs., and Supplementary Tables are available in a Source data file. The sequence of reference genome PlasmoDB-44_Pfalciparum3D7 is available at PlasmoDB (https://plasmodb.org/common/downloads/release-44/Pfalciparum3D7/fasta/data/). Source data are provided with this paper.

### Materials availability

Original DNA samples used to generate in-house WGS data are only available after discussion with corresponding authors and approval from the local investigators (at AFRIMS) who conducted the studies during which clinical samples were collected.

### Code availability

Source code for custom packages or scripts are all publicly available as GitHub repositories under the MIT license, including: (1) tskibd, the true IBD inference tool (https://github.com/bguo068/tskibd, v0.0.1); (2) ibdutils, a small python package to facilitate identity-by-descent-based analysis (https://github.com/bguo068/ibdutils, v0.1.0); (3) snp_call_nf: a Nextflow pipeline for *Plasmodium* SNP calling (https://github.com/bguo068/snp_call_nf, v0.1.0); (4) posseleff_simulations: a Nextflow pipeline to assess the impact of positive selection on Identity-by-Descent (IBD)-based inferences, utilizing population genetic simulations and true IBD methodologies (https://github.com/bguo068/posseleff_simulations, v0.1.0); (5) posseleff_empirical: a Nextflow pipeline for analyzing empirical WGS data for the effect of positive selection on IBD-based inference (https://github.com/bguo068/posseleff_empirical, v0.1.1).

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

## Acknowledgements

We would like to thank the participants in studies contributing clinical samples from which the parasite WGS data were generated, as well as the clinical investigators at the Armed Forces Research Institute of Medical Sciences who conducted the studies contributing parasite isolates to our in-house data set. This publication uses data from the MalariaGEN Consortium and *Plasmodium falciparum* Community Project as described in "An open data set of *Plasmodium falciparum* genome variation in 7000 worldwide samples. MalariaGEN et al., Wellcome Open Research 2021642 DOI: 10.12688/wellcomeopenres.16168.1." This work was supported by NIH 1R01AI145852 granted to ST-H and TDO by the U.S. National Institutes of Health.

## Author contributions

Study conception: B.G., T.D.O. and S.T.-H. Clinical sample collection: M.D.S., M.W., B.A.V. and N.C.W. Sequence data generation and processing: B.G., J.C.S., T.D.O. and S.T.-H. Data analysis: B.G., V.B. and R.L. Data interpretation: B.G., V.B., J.C.S., T.D.O. and S.T.-H. Manuscript preparation: B.G., T.D.O. and S.T.-H. All authors reviewed the final manuscript.

## Competing interests

The authors declare no competing interests.

## Inclusion and ethics

This study involved secondary analysis *P. falciparum* whole-genome sequence data. Most of the analyzed data are publicly available. New whole-genome sequence data were generated as part of a long-standing collaboration between investigators at the University of Maryland School of Medicine (UMSOM) and the Armed Forces Research Institute of Medical Sciences (AFRIMS). AFRIMS investigators conducted the clinical studies and collected the samples from which the parasite sequence data were generated and are included as authors. Sequencing was performed at UMSOM using sequencing platforms unavailable at AFRIMS. AFRIMS investigators and staff were involved in data analysis and interpretation through in-person training of staff in analytical approaches and regular virtual meetings of investigators, trainees, and staff from both institutions.
