## [Peer Review File · Nature Communications]

Strong Positive Selection Biases Identity-By-Descent-Based Inferences of Recent Demography and Population Structure in *Plasmodium falciparum*REVIEWER COMMENTS

Reviewer #1 (Remarks to the Author):

This manuscript presents a clear analysis of the effects of strong recent selection (typically mediated by drugs) on relatedness-based inference of demography and population structure in *Plasmodium falciparum*. Though the *Plasmodium* research community was quick to realize the value of IBD for detecting strong recent selection in the form of enhanced cross-population relatedness for selection-driven introgression, this manuscript demonstrates how selection can distort other inferences based on IBD metrics. The manuscript is clearly written and organized, the methods appear to be generally appropriate, and the conclusions are well supported by data and simulations. This manuscript will be of interest to the *Plasmodium* genomics community. Several analyses, such as the iterative removal of samples with strong IBD connections to reduce background and enhance relatedness peaks, are beautifully demonstrated and should be widely implemented in the field.

I have several small suggestions to improve the clarity of the manuscript:

Figure 3: It would be useful to spell out words fully in axis labels or explain abbreviations in the legend. There is room in most panels to spell out 'Neutral' instead of 'Neu', and at first glance I was trying to figure out how N_e (effective pop size) times μ (mutation rate) came into play here.

Figure 5: As with Fig 3, would be useful to have clearer axis labels, and/or more informative figure legends, to make figures more interpretable. On this figure I had to go back to the text to remind myself what C1, C2, etc. represented.

Figure 6: I do not have a clear understanding of how the red vs. gray components of the plotted bars were determined.

Reviewer #2 (Remarks to the Author):

Guo and colleagues address an important question for malaria epidemiology studies: whether a widely used approach to investigate parasite relationships (genome wide IBD) is influenced by the widespread natural selection known to occur in parasite populations as a result of on-going drug treatment and other selective forces. They do this using both simulation and by examination of empirical data. Simulation is first used to examine how IBD measures are affected in populations in which selection is present or absent. This reveals that selection results in pervasive bias, increasing the proportion of long IBD segments, enriching IBD around selected sites, resulting in overestimation of relatedness, underestimates of N_e and reduced resolution of population structure. To remedy the bias observed in simulated data with selection, they remove selected regions from the analysis, which effectively removed the bias observed. They then examine how this approach (removed selected regions from the analysis) impacts measures in natural populations. This reveals minimal impact in low transmission regions in SE Asia, but reduces bias in high transmission regions such as West Africa.

Several aspects of the paper are extremely strong:

- (i) The paper is very clearly written and explained, so is accessible to non-experts.
- (ii) The modelling is carefully conducted with use of state-of-the-art approaches for both coalescent and forward simulation. Similarly, the statistical analysis of the population genetic parameters is rigorous and carefully conducted
- (iii) The authors explore the impact of selection on a wide range of parameters that are important for work on malaria parasite genomic epidemiology and evolution
- (iv) The paper presents some practical approaches to remove bias in empirical data sets.

(v) The data and code used in provided, allowing others to repeat the analyses, and further explore the approaches used.

However, I several features of the analysis require clarification:

1. Inbreeding and recombination rates in Plasmodium: I am not convinced that the simulations account for a key aspect of Plasmodium biology: pervasive inbreeding. This occurs because most gamete fusion between males and female gametes occurs between genetically identical parasites. As a consequence, the effective per generation recombination rate is substantially lower. Inbreeding is particularly high in low transmission populations, because most infections contain a single parasite genotype, but is lower in high transmission populations where infections containing multiple clones are more common, so opportunities for outcrossing are greater.

The simulations use the recombination rates determined from Plasmodium genetic crosses. E.g. Ln 108-111: To evaluate the direct effect of positive selection on demographic inference, we conducted population genetic simulations using simplified models that reflect parameter values observed in Pf, such as strong positive selection, decreasing N_e , and a high recombination rate. Likewise the online methods (line 503-504) state: We assume a constant recombination rate of 6.67×10^7 per generation per bp (15kb/cM) 41,44.

Please can the authors clarify whether inbreeding was modelled in the simulations and if so, how was it implemented? This is important because recombination rates in Plasmodium population are often extremely low due to extensive inbreeding.

In some respects, this is a minor critique: the simulations provided clearly provide proof-of-principal that selection can bias population genetic parameters. However, this clarification would help readers to understand the magnitude of bias expected in different Plasmodium populations.

Related point. I also wonder whether modification of inbreeding coefficients could be an effective approach to simulate Plasmodium population genomics in high and low transmission regions using SLIM. The approach used for "simulation high background relatedness" (online methods 613-623), by retaining offspring from highly related parents, seems a little artificial. I wonder if this could be achieved in a more biologically realistic manner by modifying inbreeding coefficients? I'd be interested to hear the authors comments on this.

2. My expectation before reading this paper, was that removing selected regions would have a massive impact in SEA, but a lower impact in high transmission regions. So I was very surprised when you found the opposite pattern!

However, I am not sure I agree with your explanation: Line 291-293. "These findings suggest that in cases with simultaneous high background relatedness and strong positive selection, the reduction in genetic diversity and the blurring in true population structure could be dominated by high background relatedness rather than by selective sweeps."

Empirical data on low transmission SE Asian parasite populations suggest that selective sweeps driven by drug resistance are extremely broad, and often genome wide (e.g. see <https://pubmed.ncbi.nlm.nih.gov/23624527/>). This occurs because effective recombination rates are extremely low (due to inbreeding) (e.g. see <https://pubmed.ncbi.nlm.nih.gov/11018154/>). In contrast, selective sweeps have a more local impact in high transmission regions such as West Africa, where effective recombination is much higher. Hence in West Africa, removal of selected regions removes bias in IBD measures. While in SE Asia, selection has a much broader impact, so removing regions locally around selected sites has minimal impact on bias. Hence, I suspect that high relatedness per se is not the explanation. Rather, low recombination and far reaching impact of selective events, are more likely involved. Once again, incorporating inbreeding in the simulation framework could help examine this in a modelling framework.

Summary: I found this paper to be well written, stimulating and important. However, I think the paper could be improved by more explicit incorporation of inbreeding into the modelling framework, as this dramatically reduces effective recombination rates in Plasmodium populations.

RESPONSE TO REVIEWERS' COMMENTS

We thank the reviewers for their time and effort in reviewing our manuscript and for their valuable feedback that has greatly improved the quality of the manuscript. Here, we address the reviewers' comments point-by-point as detailed below in **blue** font.

Reviewer #1 (Remarks to the Author):

This manuscript presents a clear analysis of the effects of strong recent selection (typically mediated by drugs) on relatedness-based inference of demography and population structure in *Plasmodium falciparum*. Though the *Plasmodium* research community was quick to realize the value of IBD for detecting strong recent selection in the form of enhanced cross-population relatedness for selection-driven introgression, this manuscript demonstrates how selection can distort other inferences based on IBD metrics. The manuscript is clearly written and organized, the methods appear to be generally appropriate, and the conclusions are well supported by data and simulations. This manuscript will be of interest to the *Plasmodium* genomics community. Several analyses, such as the iterative removal of samples with strong IBD connections to reduce background and enhance relatedness peaks, are beautifully demonstrated and should be widely implemented in the field. I have several small suggestions to improve the clarity of the manuscript:

REVIEWER COMMENT

Figure 3: It would be useful to spell out words fully in axis labels or explain abbreviations in the legend. There is room in most panels to spell out 'Neutral' instead of 'Neu', and at first glance, I was trying to figure out how N_e (effective pop size) times μ (mutation rate) came into play here.

AUTHOR RESPONSE

We have revised Fig 3, as well as all other relevant main and supplementary figures and tables to ensure clarity. Specifically, we have made the following changes: 1) expanded 'Neu' to 'Neutral' in all instances where it appeared, and 2) replaced 'Sel' with 'Selection'.

REVIEWER COMMENT

Figure 5: As with Fig 3, it would be useful to have clearer axis labels, and/or more informative figure legends, to make figures more interpretable. On this figure, I had to go back to the text to remind myself what C1, C2, etc. represented.

AUTHOR RESPONSE

Per the reviewer's suggestion, we have revised the axis labels in Fig 3F and Fig 5C to read "detected community labels" instead of "detected population label." This change clarifies that the 'C' in the tick labels refers to different communities. In the figure legends for both Fig 3F and Fig 5C, we have also added detailed explanations of the 'C0-C4' labels. For instance, in the legend for Fig 3F, we now specify: "For each subplot, columns represent the largest five detected communities, labeled as C0 to C4, with C0 being the largest and C4 the fifth largest." This addition aims to provide immediate context to the reader without the need for cross-referencing with the main text.

REVIEWER COMMENT

Figure 6: I do not have a clear understanding of how the red vs. gray components of the plotted bars were determined.

AUTHOR RESPONSE

As suggested by the reviewer, we have updated the legends for Fig 6b-c with a more detailed description of the color coding. In Fig 6b, we illustrate the distribution of community sizes detected in IBD networks using the West Africa (WAF) data set before any modification. Here, the leftmost red bar labeled C0 represents the

dominant community in the original IBD network, with a size of 1,222 isolates. Fig 6c, on the other hand, shows the distribution after removing IBD peaks. This process leads to a reassignment of isolates from the dominant community (C0 in Fig 6b) into smaller, distinct communities, now labeled as C0-C14 in Fig 6c. To visually convey how community assignments have shifted as a result of this reassignment, each bar in Fig 6c is split into two color components: red and gray. The red portion represents isolates that were part of the original dominant community (community C0 in Fig 6b), while the gray portion indicates isolates that are not from this dominant community. We hope this updated explanation provides a clearer understanding of the color scheme used in Fig 6 and its significance in illustrating the impact of removing IBD peaks on community distribution in the data set.

Reviewer #2 (Remarks to the Author):

Guo and colleagues address an important question for malaria epidemiology studies: whether a widely used approach to investigate parasite relationships (genome-wide IBD) is influenced by the widespread natural selection known to occur in parasite populations as a result of ongoing drug treatment and other selective forces. They do this using both simulation and examination of empirical data. Simulation is first used to examine how IBD measures are affected in populations in which selection is present or absent. This reveals that selection results in pervasive bias, increasing the proportion of long IBD segments, enriching IBD around selected sites, resulting in overestimation of relatedness, underestimates of N_e and reduced resolution of population structure. To remedy the bias observed in simulated data with selection, they remove selected regions from the analysis, which effectively removes the bias observed. They then examine how this approach (removing selected regions from the analysis) impacts measures in natural populations. This reveals minimal impact in low transmission regions in SE Asia, but reduces bias in high transmission regions such as West Africa.

Several aspects of the paper are extremely strong:

- (i) The paper is very clearly written and explained, so is accessible to non-experts.
- (ii) The modeling is carefully conducted with use of state-of-the-art approaches for both coalescent and forward simulation. Similarly, the statistical analysis of the population's genetic parameters is rigorous and carefully conducted
- (iii) The authors explore the impact of selection on a wide range of parameters that are important for work on malaria parasite genomic epidemiology and evolution
- (iv) The paper presents some practical approaches to remove bias in empirical data sets.
- (v) The data and code used is provided, allowing others to repeat the analyses, and further explore the approaches used.

However, several features of the analysis require clarification:

REVIEWER COMMENT

Inbreeding and recombination rates in *Plasmodium*: I am not convinced that the simulations account for a key aspect of *Plasmodium* biology: pervasive inbreeding. This occurs because most gamete fusion between male and female gametes occurs between genetically identical parasites. As a consequence, the effective per-generation recombination rate is substantially lower. Inbreeding is particularly high in low transmission populations, because most infections contain a single parasite genotype, but is lower in high transmission populations where infections containing multiple clones are more common, so opportunities for outcrossing are greater.

The simulations use the recombination rates determined from *Plasmodium* genetic crosses. E.g. Ln 108-111: To evaluate the direct effect of positive selection on demographic inference, we conducted population genetic simulations using simplified models that reflect parameter values observed in Pf, such as strong positive selection, decreasing N_e , and a high recombination rate. Likewise the online methods (line 503-504) state: We assume a constant recombination rate of 6.67×10^7 per generation per bp (15kb/cM) 41,44.

Please can the authors clarify whether inbreeding was modeled in the simulations and if so, how was it implemented? This is important because recombination rates in the *Plasmodium* population are often extremely low due to extensive inbreeding.

AUTHOR RESPONSE

Thank you for your insightful comments regarding the modeling of inbreeding and recombination rates in *P. falciparum* (*Pf*). We appreciate the opportunity to clarify these aspects in our simulations.

Differentiating recombination rates: We recognize the importance of distinguishing between the true (i.e., specified parameter) recombination rate and the effective recombination rate in *Pf* genomes. In our simulations, the recombination rate parameter (6.67×10^{-7} per generation per bp, or 15kb/cM) refers to the *true* recombination rate, which accounts for all recombination events, including both *effective* recombination events and those that are *unobservable* due to genetic identity between recombining genomes, consistent with the concept discussed in Camponovo *et al.* (2022). This distinction aligns with observations in empirical data where *high* inbreeding can lead to many recombination events being effectively *invisible* (low effective recombination rate). In our simulations, the effective recombination rate *cannot* be directly specified but can be approached by setting the true recombination rate to 6.67×10^{-7} per generation per bp and adjusting other parameters that control the level of inbreeding. We have provided a detailed comparison of true (parameter) versus effective recombination rates and a description of inbreeding modeling methods in the newly added Supplementary Notes, Section 2.

Inbreeding modeled in our original manuscript: In the original manuscript, inbreeding is incorporated mainly via two means, including (1) shrinking the population size while keeping mating random (in the single population model), and (2) promoting assortative mating in which a parent tends to mate with another parent with a high relatedness (can be triggered in both single- and multi-population models).

(1) In the single population model, representative of low transmission settings such as SEA, inbreeding was incorporated via shrinking population size in recent history (described in Online Methods of the original manuscript; now available in Supplementary Notes). In the neutral case, the effective population size (N_e) closely follows the parameter (census) population size, N , given random mating; in the presence of selection, N_e estimates are biased by selection which leads to an even smaller estimated N_e . In both cases, we have a small N_e , especially for the recent generations. The inbreeding coefficient is usually defined as the probability that two alleles at a locus in a diploid individual are identical-by-descent (Walsh & Lynch, 2018). In the case of *Pf*, our genome data represents haploid parasite stages sampled from the human host. We use the genome-wide average IBD sharing proportion as a proxy for measuring the level of inbreeding for these *haploid* genomes. With a small effective population size, the probability of two alleles at the same locus of a random pair of haploid genomes being IBD is high and thus *inbreeding potential* is also higher. We defined and reasoned the metrics of inbreeding potential as a proxy of inbreeding levels in haploid genomes in the Supplementary Notes, Section 1.1.

(2) The second method used in our original manuscript to simulate high relatedness (inbreeding) employed a type of assortative mating where closer relatives tend to mate more frequently than distant relatives. This type of inbreeding may be present in intermediate/low transmission settings where the parasite population is more structured and mating is more frequent between parasites that are in closer proximity and likely to be highly genetically related. We implemented this type of inbreeding following the SLiM documentation on simulating relatedness via one of the two callbacks, *modifyChild* and *mateChoice*. In case more technical details are interested, we provide the SLiM script in our GitHub repository (https://github.com/bquo068/posseleff_simulations/tree/main/slim).

Improvements on inbreeding modeling: In this revision, we have improved our inbreeding modeling in the following ways: (1) We have expanded the shrinking population size methods to both single/multiple populations and analyzed how different present-day population sizes (i.e. different extents of inbreeding due to population size shrinking) affect the bias resulting from selection and effects of selection correction; (2) We updated the assortative mating strategy so that the inbreeding modeling is highly parameterized and “tunable”

(as detailed in Supplementary Notes, Section 2.4.2) to evaluate how different levels of inbreeding affect selection bias; (3) We also added a third inbreeding modeling strategy via selfing to cover a special case of non-random mating. We discuss the simulation methods based on all three types of inbreeding models below and in the Supplementary Notes, Section 2.4.

REVIEWER COMMENT

In some respects, this is a minor critique: the simulations clearly provide proof-of-principal that selection can bias population genetic parameters. However, this clarification would help readers to understand the magnitude of bias expected in different *Plasmodium* populations.

AUTHOR RESPONSE

We agree with the reviewer regarding the benefits of quantifying the magnitude of bias in different *Plasmodium* populations, and how this value might indicate the need for and effectiveness of bias corrections. Although inbreeding is complicated and many relevant parameters are unknown, we have tried to approach the question by making some assumptions about the model and parameters and exploring how different types and levels of inbreeding can affect the magnitude of bias and the need for correction. Detailed responses and a description of additional analyses related to this aspect are integrated with the response to the comment below, where we elaborate on the specific methodologies and findings of our revised simulations.

REVIEWER COMMENT

Related point. I also wonder whether modification of inbreeding coefficients could be an effective approach to simulate *Plasmodium* population genomics in high and low transmission regions using SLiM. The approach used for “simulation of high background relatedness” (online methods 613-623), by retaining offspring from highly related parents, seems a little artificial. I wonder if this could be achieved in a more biologically realistic manner by modifying inbreeding coefficients. I’d be interested to hear the author’s comments on this.

AUTHOR RESPONSE:

We concur that adopting a biologically realistic approach to control the inbreeding coefficient will significantly enhance our understanding of selection bias in parasite demography estimations. Specifically, by simulating selection in populations with varying inbreeding coefficients, we can better ascertain at which inbreeding levels selection bias becomes negligible or requires correction.

We have simulated a spectrum of inbreeding via three distinct strategies and performed a detailed analysis of how different types and levels of inbreeding affect selection bias (see Supplementary Notes). Here, we highlight a few important points from these additional inbreeding simulations in response to the reviewer’s comments.

We used simplified inbreeding simulation models: Modeling inbreeding can be extremely difficult given the complex life cycle of *P. falciparum*. Realistic inbreeding modeling would require multiple levels of control: inbreeding within mosquitos, inbreeding due to the division of the oocyst population into individual mosquitoes, and, inbreeding due to geographic isolation (Anderson *et al.* 2000). Inbreeding simulation with this level of complexity is beyond the scope of our current study. Here, we modeled inbreeding with simplified assumptions: we assumed a Wright-Fisher population of hermaphrodite diploids, with two haploid genomes corresponding to each present-day diploid. The simplified model may be somewhat artificial, as indicated by the reviewer; however, it can still be utilized to explore how different levels of inbreeding influenced the effect of strong positive selection on IBD-based inferences of demography and population structure. (This point is also presented in Supplementary Notes, Section 2.1)

The inbreeding coefficient cannot be directly specified but can be indirectly modified: Although SLiM does not offer direct control over the inbreeding coefficient, in our additional analyses, we effectively managed inbreeding through parameters influencing assortative mating, selfing, and population size changes, as noted above and described in more detail below. Our strategy involved adjusting these parameters based on post-

simulation inbreeding potential assessments, using inbreeding potential estimates (defined in Supplementary Notes, Section 1.1) to guide us toward desired inbreeding levels. (Also see Supplementary Notes, Section 2.3)

Three strategies were used to modify inbreeding levels: We modeled inbreeding via three different strategies in both the single- and multiple-population models, to provide a more comprehensive understanding of how different levels and types of inbreeding affect the magnitude of selection bias (Supplementary Notes, Section 2.4). The three strategies include: (1) shrinking population size so that haploid genomes have high genetic relatedness on average, (2) positive assortative mating such that parasites tend to mate with close relatives, and (3) selfing, the mating of male and female gametes from the same individual (diploid zygotes in mosquitoes) to form a zygote for the offspring generation. We describe how each of the inbreeding simulations was implemented in the Supplementary Notes, Section 2.4. For each strategy, we varied the value of a key parameter such that a spectrum of inbreeding levels was simulated. By modeling inbreeding using distinct strategies, we hope to understand the robustness of our results to different inbreeding mechanisms, each of which might be a part of a more complex and biologically realistic model of inbreeding for *Pf*.

Quantitative metrics for measuring local/global impact of selective sweeps: Depending on the level of inbreeding, strong positive selection can affect the chromosome locally (in low inbreeding populations) and globally (in high inbreeding populations). We proposed the terms “peak impact index” and “global impact index” for measuring the genomically local or global impact of selective sweeps. The peak impact index is essentially the amount of increased IBD contributed by a peak relative to the baseline level of IBD. This metric takes into account both the peak width and height and the background relatedness/IBD/inbreeding level. As discussed below, the peak impact index can help determine whether the peak-removal selection bias correction is necessary. The global impact index, on the other hand, represents selection-induced changes that have affected the whole chromosome, which has little local bias and is not corrected by removing IBD peaks. The definitions of these terms are detailed in Supplementary Notes, Sections 1.2 and 1.4.

A wide range of inbreeding levels can be simulated by varying the value of a key parameter: By decreasing the value of parameter N_0 of the shrinking-population-size model, decreasing parameter δ of the assortative mating model, and increasing the *selfing rate* of the selfing-based inbreeding model, we covered in these simulations a wide spectrum of inbreeding potential values from 0.0005 - 0.33, inclusive of values representing the *Pf* populations in SEA (0.0157) and WAF (0.0005). Please see the detailed analysis results in the Supplementary Notes, Sections 3.1-3.3 and Figs. S11-S16.

Modifying the inbreeding level affects the magnitude of genomically local selection bias: From inbreeding simulations modeled via shrinking population size (Fig S11-S12) and positive assortative mating (Fig S13-S14), we found that with increased levels of inbreeding, selective sweeps show a decreased local impact/bias (as measured by the *peak impact index*). In these simulations, the inbreeding level tends to be negatively correlated with local impact (Fig S17, left two columns), and thus, the removal of IBD peaks has less of an impact and may not be necessary in high inbreeding populations. The pattern is different when inbreeding is modeled via selfing, where local impact first increases and then decreases with increasing inbreeding levels (Fig. S15-S16, and S17 last column), likely due to a synergistic effect of selfing-mediated inbreeding on selection (faster increase in frequency of selected alleles for high selfing rates, Fig S15c). However, except for some extreme cases, the peak impact index can still serve as a useful metric to quantify the magnitude of (local) selection bias, under all three modeling scenarios. We hope the extended inbreeding modeling and quantitative analysis of selection bias provide a deeper insight into how different types and levels of inbreeding affect the magnitude of selection bias.

REVIEWER COMMENT

My expectation before reading this paper was that removing selected regions would have a massive impact in SEA, but a lower impact in high-transmission regions. So I was very surprised when you found the opposite pattern!

However, I am not sure I agree with your explanation: Line 291-293. “These findings suggest that in cases with simultaneous high background relatedness and strong positive selection, the reduction in genetic diversity and

the blurring in true population structure could be dominated by high background relatedness rather than by selective sweeps.”

Empirical data on low transmission SE Asian parasite populations suggest that selective sweeps driven by drug resistance are extremely broad, and often genome-wide (e.g. see <https://pubmed.ncbi.nlm.nih.gov/23624527/>). This occurs because effective recombination rates are extremely low (due to inbreeding) (e.g. see <https://pubmed.ncbi.nlm.nih.gov/11018154/>). In contrast, selective sweeps have a more local impact in high transmission regions such as West Africa, where effective recombination is much higher. Hence in West Africa, removal of selected regions removes bias in IBD measures. While in SE Asia, selection has a much broader impact, so removing regions locally around selected sites has minimal impact on bias. Hence, I suspect that high relatedness per se is not the explanation. Rather, low recombination and far-reaching impact of selective events are more likely involved. Once again, incorporating inbreeding in the simulation framework could help examine this in a modeling framework.

AUTHOR RESPONSE:

We are again grateful for these valuable insights and acknowledge the distinction between the possible explanations for our findings. Using the framework described above, we conducted a further integrated analysis to *quantitatively* examine the relationship between inbreeding and the local/global impacts of selective sweeps. Our findings, summarized in Fig. S17, show that increased inbreeding tends to diminish the local impact of selective sweeps (that is correctable) but correlates with a more pronounced global impact (uncorrectable or unnecessary to correct). An interesting divergence occurs when inbreeding is modeled via selfing. In these cases (selfing), both local and global impacts intensify as inbreeding increases, potentially due to the synergistic effects of selfing and selection. This observation suggests that the interplay between inbreeding and selection is complex and may require further expanded simulation analyses for a comprehensive understanding. In general, our results support the hypothesis that higher levels of inbreeding and lower effective recombination are associated with a higher global impact of positive selection. The efficacy of peak-removal-based correction tends to be negatively related to the level of inbreeding and the magnitude of global impact, and positively correlated with local impact (peak impact index).

These results are largely consistent with the reviewer's suggested explanation: (1) in high transmission regions such as WAF, where the effective recombination rate is high, the global impact is small and local impact (peak impact index) is relatively large, local selection bias in IBD can be corrected; (2) in low transmission regions such as SEA, where inbreeding is high and the effective recombination rate is low, the global impact of selective sweeps is large, and local impact is relatively small which makes selection bias either uncorrectable or unnecessary to correct. We have updated the related paragraphs in both the results and discussion sections to reflect a more balanced and evidence-based explanation.

Summary: I found this paper to be well-written, stimulating, and important. However, I think the paper could be improved by more explicit incorporation of inbreeding into the modeling framework, as this dramatically reduces effective recombination rates in *Plasmodium* populations.

AUTHOR RESPONSE

We thank the reviewers for their constructive suggestions that have improved the quality and utility of our work.

References:

- T. J. C. Anderson, R. E. L. Paul, C. A. Donnelly, K. P. Day, Do malaria parasites mate non-randomly in the mosquito midgut? *Genetics Research* **75**, 285–296 (2000).
- F. Camponovo, C. O. Buckee, A. R. Taylor, Measurably recombining malaria parasites. *Trends in Parasitology*, doi: 10.1016/j.pt.2022.11.002 (2022).
- B. Walsh, M. Lynch, *Evolution and Selection of Quantitative Traits* (Oxford University Press, New York, NY, 2018).

REVIEWERS' COMMENTS

Reviewer #2 (Remarks to the Author):

The authors have done an extremely thorough job revising the manuscript after the initial reviews. I have no further comments. This is an important paper that identifies some clear biases resulting from use of IBD based inferences, and evaluates some simple practical solutions to reduce levels of bias.